# Hematological Parameters as Potential Markers for Distinguishing Pulmonary Tuberculosis from Genitourinary Tuberculosis

**DOI:** 10.3390/pathogens12010084

**Published:** 2023-01-04

**Authors:** Hui-Zin Tu, Tsung-Jen Lai, Yao-Shen Chen, Herng-Sheng Lee, Jin-Shuen Chen

**Affiliations:** 1Division of Clinical Laboratory, Department of Pathology and Laboratory Medicine, Kaohsiung Veterans General Hospital, No.386, Dazhong 1st Rd., Zuoying Dist., Kaohsiung 813414, Taiwan; 2Department of Internal Medicine, Kaohsiung Veterans General Hospital, No.386, Dazhong 1st Rd., Zuoying Dist., Kaohsiung 813414, Taiwan; 3Department of Pathology and Laboratory Medicine, Kaohsiung Veterans General Hospital, No.386, Dazhong 1st Rd., Zuoying Dist., Kaohsiung 813414, Taiwan; 4Division of Nephrology, Department of Internal Medicine, Kaohsiung Veterans General Hospital, Dazhong 1st Rd., Zuoying Dist., Kaohsiung 813414, Taiwan

**Keywords:** *Mycobacterium tuberculosis* complex, genitourinary tuberculosis, hematological tests, receiver operating characteristic

## Abstract

*Mycobacterium tuberculosis* complex (MTBC) infection is an important public health concern in Taiwan. In addition to pulmonary tuberculosis (PTB), MTBC can also cause genitourinary tuberculosis (GUTB). This study aimed to examine the role of laboratory data and the values that can be calculated from them for the early detection of GUTB. Patients admitted from 2011 to 2020 were retrospectively recruited to analyze their associated clinical data. Statistical significance was analyzed using the chi-square test and univariate analysis for different variables. A receiver operating characteristic (ROC) curve analysis was used to evaluate the performances of the examined laboratory data and their calculated items, including the neutrophil-to-lymphocyte ratio (NLR), monocyte-to-lymphocyte ratio (MLR), neutrophil-to-monocyte-plus-lymphocyte ratio (NMLR), and platelet-to-lymphocyte ratio (PLR), in diagnosing PTB or GUTB. A *p*-value of <0.05 was considered significant. The ROC curve showed that the discriminative power of the neutrophil count, NLR, and MLR was within the acceptable level between patients with both PTB and GUTB and those with GUTB alone (area under the curve [AUC] values = 0.738, 0.779, and 0.725; *p* = 0.024, 0.008, and 0.033, respectively). The discriminative power of monocytes and the MLR was within the acceptable level (AUC = 0.782 and 0.778; *p* = 0.008 and 0.010, respectively). Meanwhile, the neutrophil and lymphocyte counts, NLR, NMLR, and PLR had good discriminative power (AUC = 0.916, 0.896, 0.898, 0.920, and 0.800; *p* < 0.001, <0.001, <0.001, <0.001, and 0.005, respectively) between patients with GUTB and those with PTB alone. In conclusion, the neutrophil count, lymphocyte count, NLR, NMLR, and PLR can be used as potential markers for distinguishing PTB from GUTB.

## 1. Introduction

*Mycobacterium tuberculosis* complex (MTBC) is one of the leading infectious agents in Taiwan. In 2019, there were 8732 cases (37 cases per 100,000 population) of tuberculosis (TB) reported in Taiwan [1]. Although TB incidence is starting to decline after the implementation of several efforts to control this disease, it remains an important public health concern.

MTBC usually causes pulmonary TB (PTB); however, it can also spread from the lungs to other organs via the lymphatic system or the bloodstream causing extrapulmonary tuberculosis (EPTB) [2]. Among the forms of EPTB, genitourinary tuberculosis (GUTB) is the second most common (20–40%) in developing countries and third most common in developed countries [3]. GUTB is usually caused by the spread of MTBC in the bloodstream during the initial stage of infection [4]. The diagnosis of GUTB is established when MTBC is detected in the urine. In addition to medium culture, other symptoms and signs, including dysuria, sterile pyuria, hematuria, and characteristic radiographic findings, are helpful in diagnosing this condition [5]. Early diagnosis of tuberculosis using these tests could help control transmission of the pathogen. Currently, bacterial culture and histological examination are used to diagnose tuberculosis. However, these laboratory tests often take more than three weeks. Therefore, identifying faster reference indicators has become an important issue for disease control.

Some studies have proposed the use of basic laboratory measures, such as peripheral blood tests, to predict the risk of TB [6,7,8,9,10]. The neutrophil-to-lymphocyte ratio (NLR), monocyte-to-lymphocyte ratio (MLR), and platelet-to-lymphocyte ratio (PLR) may be utilized in the detection of incident symptomatic TB [6]. When a cut-off value of less than 7.0 was used, the sensitivity of NLR for predicting TB was 91% and the specificity was 82% [7]. The NLR’s receiver operating characteristic (ROC) may perform better than C-reactive protein (CRP) in diagnosing TB [8]. The detection values of the white blood cell (WBC) count, neutrophil count, MLR, PLR, systemic immune-inflammation index, and procalcitonin levels may decrease significantly following TB therapy [9].The mechanism of MTBC-activated monocytes may involve cytokine stimulation by CD4+ T cells [10]. In addition, some studies have demonstrated that changes in some biochemical parameters, such as the levels of serum sodium (Na), albumin, calcium (Ca), and potassium (K), were associated with TB itself or the treatment effect of TB [11,12].

However, the correlation study did not explore the relationship between the laboratory data and GUTB. Therefore, we aimed to investigate the correlation between these two variables using our database and to characterize the early diagnosis of GUTB using laboratory data.

## 2. Materials and Methods

### 2.1. Data Sources

This retrospective study was conducted at Kaohsiung Veterans General Hospital (KSVGH). All de-identified patient data, including patient diagnosis, treatment, and laboratory test results, were collected from the clinical database of KSVGH. The data collected during the initial medical examination included information on each patient’s sex, age, laboratory findings, and history of comorbidities (e.g., chronic kidney disease (CKD), hypertension, diabetes mellitus (DM), ischemic heart disease (IHD), chronic obstructive pulmonary disease (COPD), liver cirrhosis, malignancy, and acquired immune deficiency syndrome (AIDS)). The comorbidities were defined based on the International Classification of Diseases (ICD) (9th and 10th revisions) coding system. This study was approved by the Institutional Review Board of the KSVGH (KSVGH21-CT1-16).

A flowchart of the recruitment process is presented in Figure 1. A total of 1723 patients who underwent sputum and urine culture tests for the detection of MTBC in KSVGH from 1 January 2011 to 31 December 2020 were recruited. Patients (1) who received Bacillus Calmette-Guerin (BCG) treatment for urinary bladder cancer,(2) with incomplete data on TB culture and biochemical and hematological tests, and (3) whose diagnosis and culture results did not match were excluded. Patient diagnoses were made based on the ICD-9 and ICD-10 coding systems. Subsequently, 234 patients were excluded: 5 patients who received BCG treatment, 17 patients with missing data, and 212 unmatched patients. Based on the MTBC culture results, 1489 patients were categorized into four groups (A–D) as follows:(A) 69 patients with positive sputum culture and urine culture tests [labeled as Sputum+&Urine+], (B) 16 patients with negative sputum culture but positive urine culture tests [labeled as Sputum−&Urine+], (C) 110 patients with positive sputum culture but negative urine culture tests [labeled as Sputum+&Urine−], and (D) 1294 patients with negative sputum culture and negative urine culture tests [labeled as Sputum−&Urine−].

### 2.2. Biochemical and Hematological Data

The data collected during the initial medical examination included demographic characteristics, sex, and age. Data on the following biochemical parameters were also obtained: glucose, CRP, creatinine, blood urea nitrogen (BUN), the estimated glomerular filtration rate (EGFR), uric acid, glutamic oxaloacetic transaminase (GOT), glutamic pyruvic transaminase (GPT), total bilirubin, lactate dehydrogenase (LDH), albumin, triglyceride (TG), cholesterol, high-density lipoprotein (HDL), low-density lipoprotein (LDL), sodium (Na), potassium (K), and calcium (Ca). Moreover, data on the following hematological parameters were collected: red blood cell (RBC), WBC, platelet, neutrophil, lymphocyte, monocyte, eosinophil, and basophil counts; hemoglobin (Hb) levels; and hematocrit (Hct) values. After that, the NLR, MLR, neutrophil-to-monocyte-plus-lymphocyte ratio (NMLR), and PLR were determined. The NLR was calculated as the number of neutrophils divided by the number of lymphocytes, the MLR as the number of monocytes divided by the number of lymphocytes, the NMLR as the number of neutrophils divided by the number of lymphocytes and monocytes, and the PLR as the number of platelets divided by the number of lymphocytes. Data on all indicators were collected within three months prior to the patient receiving a diagnosis. After data cleaning, we excluded variables with a data loss rate of more than 5%. For the retained variables, missing values were replaced using SPSS.

### 2.3. Statistical Analysis

Before analysis, all groups were paired by probability score matching (PSM). Statistical significance was analyzed using a chi-square test for categorical variables. A univariate analysis was performed to identify the variables associated with GUTB. Receiver operating characteristic (ROC) curve analysis was used to evaluate the performances of the neutrophil, lymphocyte, monocyte, and eosinophil counts; NLR; MLR; NMLR; and PLR in diagnosing PTB or GUTB. A *p*-value of <0.05 was considered significant. The probability was also estimated based on the area under the curve (AUC) of the ROC, which represents the degree or measure of separability. An AUC of >0.9 was considered outstanding, 0.8–0.9 was considered excellent, 0.7–0.8 was considered acceptable, 0.5–0.7 was considered not good, and 0.5 suggested no distinction. The data were analyzed using the Statistical Package for the Social Sciences (SPSS ver.22, Chicago, IL, USA).

## 3. Results

The demographic and clinical outcomes of the four groups [group A: Sputum+&Urine+, group B: Sputum−&Urine+, group C: Sputum+&Urine−, and group D: Sputum−&Urine−] are presented in Table 1. The proportion of men in group A was significantly higher than in groups B and D (*p* = 0.010 and 0.025). In terms of comorbidities, compared to Group D, group C had a significantly lower incidence of hypertension (*p* = 0.018), malignancy (*p* = 0.047), and AIDS (*p* = 0.013), while group A had a significantly lower incidence of DM (*p* = 0.001). These significantly different variables were paired in the subsequent analysis.

After propensity score matching (PSM), the results of the Student’s *t* test analysis of the paired laboratory data are presented in Table 2, Table 3 and Table 4. When comparing group A Sputum+&Urine+ and group B Sputum−&Urine+ (Table 2), the average neutrophil rate of group A was significantly higher than that of group B (*p* = 0.032), while the lymphocyte rate was significantly lower (*p* = 0.006). When comparing group B Sputum−&Urine+ and group C Sputum+&Urine−, the glucose level, neutrophil rate, NLR, and NMLR of group B were significantly lower than those of group C (*p* = 0.006, <0.001, <0.024, and <0.001, respectively),while the lymphocyte and monocyte rates were significantly higher (*p* < 0.001 and *p* = 0.007, respectively). When comparing group B Sputum−&Urine+ and group D Sputum−&Urine−, the glucose level of group B was significantly lower than that of group D (*p* < 0.001), while the RBC count, Hb, and Hct values were significantly higher (*p* < 0.001, *p* = 0.028, and 0.024, respectively).

When comparing group A Sputum+&Urine+ and group C Sputum+&Urine− (Table 3), the glucose level, WBC count, and platelet count of group A were significantly lower than those of group C (*p* = 0.007, 0.023, and 0.039, respectively), while the lymphocyte rate was significantly higher (*p* = 0.025). When comparing group A Sputum+&Urine+ and group D Sputum−&Urine−, the WBC counts and lymphocyte rate of group A were significantly lower than those of group D (*p* = 0.004 and 0.005, respectively), while the neutrophil rate was significantly higher (*p* = 0.006).

When comparing group C Sputum+&Urine− and group D Sputum−&Urine− (Table 4), the WBC count of group C was significantly lower than that of group D (*p* = 0.012), while the RBC, platelet counts, and neutrophil rate were significantly higher (*p* = 0.009 and 0.001, and 0.038, respectively). Results of the ROC curve analysis are presented in Figure 2 and Table 5. When comparing group A Sputum+&Urine+ and group B Sputum−&Urine+ (Table 5), the discriminative power levels of the neutrophil count, NLR, and MLR were considered acceptable (AUC values = 0.738, 0.779, and 0.725, *p* = 0.024, 0.008, and 0.033, respectively). When comparing group B Sputum−&Urine+ and group C Sputum+&Urine− (Table 5), the discriminative power levels of the monocyte count and MLR were considered acceptable (AUC values = 0.782 and 0.778, *p* = 0.008 and 0.010, respectively). The discriminative power levels of the neutrophil count, lymphocyte count, NLR, NMLR, and PLR were considered excellent (AUC = 0.916, 0.896, 0.898, 0.920, and 0.800, *p* < 0.001, <0.001, <0.001, <0.001, and 0.005, respectively).

Based on Table 5, the neutrophil count, lymphocyte count, NLR, NMLR, and PLR were better markers than the other hematological parameters (AUC = 0.8 and *p* < 0.05). Subsequently, the cut-off value was selected using the maximum Youden’s index (sensitivity + specificity-1). The stratification of the five hematological markers based on the cut-off value is shown in Table 6. If the neutrophil count was <72, lymphocyte count was >9.5, NLR was <9.6, NMLR was <2.9, or PLR was <293.3, the probability of being diagnosed with GUTB was significantly higher than the probability of being diagnosed with PTB.

## 4. Discussion

Many researchers have demonstrated that hematological data can potentially be used to distinguish PTB from GUTB. The NLR obtained at the initial diagnostic stage is a useful laboratory marker for distinguishing patients with pulmonary TB from patients without pulmonary TB. For example, an NLR of <7 was used as the optimal cut-off value to distinguish patients with pulmonary TB from those with bacterial community-acquired pneumonia [9]. The NLR also may be used as a marker of inflammation to help in the clinical management of TB patients and to determine the disease’s severity. In one study, the NLR (4.7 versus 3.1) values were higher in the group with advanced PTB than in the group with mild to moderate PTB. It is helpful to determine the severity of inflammation in patients with PTB [13]. The NMLR is a powerful marker that can be used to distinguish TB from non-TB infectious lung diseases. One study that use dan NMLR cut-off value of 1.77 to distinguish TB patients from healthy individuals found a sensitivity of 79.1% and a specificity of 82.7% [14]. These findings were consistent with our results. In addition, we evaluated whether the results of biochemical and hematological tests could discriminate PTB from GUTB.

The results of our study show that the percentage of neutrophils was significantly higher in patients with PTB than in those with GUTB. However, the percentage of lymphocytes showed a different trend. The percentage of neutrophils in patients with GUTB was significantly higher than in patients with PTB alone. Previous studies indicate that neutrophils play a significant role in the TB inflammatory process [15]. Moreover, neutrophils are associated with an increased risk of cavity formation and lung tissue damage [16]. In one study, the neutrophil counts and NLR decreased from base line following a six-month anti-TB drug therapy [17]. These results prove that neutrophils are active during TB infection and correlate with PTB. Our results showed that neutrophils were more active in PTB patients than in GUTB patients. This may be due to the different immune mechanisms involved in PTB compared to GUTB. Previous research also pointed out significant differences in correlation profiles between PTB and EPTB [18].

Based on the results of the ROC analysis, hematological data cannot be used to effectively discriminate TB from non-TB diseases. This result is different from previous studies because of differences in the control groups used. Previous studies used healthy individuals as a control group, while our control group (group D) comprised patients suspected to have TB and admitted to the hospital. Hence, blood and urine culture tests were performed. Although the results of both tests were negative, our control group patients had other pathological manifestations such that the results of their biochemical and hematological tests differed from those of the healthy population. Compared with the other three groups, no difference was observed in the biochemical or hematological data of our control group, which is further indicative of their pathological status. In addition, most patients had several existing comorbidities before being diagnosed with TB. Therefore, the test values of these patients were usually pathological. Hence, it was difficult to distinguish TB-infected patients from the control patients using the test data. However, the hematology test data are better for distinguishing the differences between PTB, GUTB, and PTB plus GUTB. Among the different hematological parameters, the ability of the neutrophil count, lymphocyte count, and NLR to discriminate GUTB from PTB was better than that of other variables. This result is consistent with that of the univariate analysis. Although the hematology data have a good discriminative ability in differentiating PTB from GUTB, the culture tests must also yield accurate results. Therefore, it is still impossible to replace the gold standard, TB culture; however, hematology data may be used as a reference for making an auxiliary diagnosis.

When infected, the immune cells in the peripheral blood get activated to increase or decrease the proportion of various types of WBCs. Therefore, the complete blood count/ differential count (CBC/DC), NLR, and MLR, etc. are regarded as rapid and simple markers of the immune system’s response to stress [19]. The acquired immunity to TB infection is cell-mediated. After the macrophages ingest the MTBC, the bacterial antigens are presented by class II major histocompatibility complex molecules. Thereafter, the CD4 T cells transfer the response signal [20,21]. The lymphocyte count plays an important role in the immune response against TB. The human immunodeficiency virus-mediated loss of CD4 T cells renders patients susceptible to TB [22]. A previous study reported that the peripheral blood lymphocyte count increased during the initial stage of infection. As the disease progressed, the infected parts, such as the pleural fluid, or the occurrence of ascites were reported to increase the number of activated lymphocytes [20]. Based on our data, the lymphocyte ratios of patients, from high to low, were as follows: Sputum−&Urine+ > Sputum+&Urine+ > Sputum+&Urine−. The NLR of Sputum−&Urine+ patients was significantly lower than that of Sputum+&Urine− patients, while the monocyte ratio of Sputum−&Urine+ patients was significantly higher than that of Sputum+&Urine− patients. The increased lymphocyte and monocyte ratio indicates that the cell-mediated acquired immunity was maintained in the Sputum−&Urine+ patients.

Further infection in patients with PTB causes EPTB, including GUTB. Therefore, these patients may exhibit a prolonged immune response that causes the lymphocyte and neutrophil counts to increase or decrease. In other words, the initial TB infection often starts from PTB. The patient’s neutrophil count is higher than that of other immune cells during this period. However, in patients with poor immunity or other viral infections such as HIV, the lymphocyte rate increased, and the neutrophil rate showed a relative decrease. TB cannot be cleared and tends to be EPTB, such as PTB with GUTB or GUTB only. Therefore, in this study, patients with only PTB had the highest NLR, while patients with PTB and GUTB or patients with GUTB only had a relatively low NLR. These results mean that the proportion of lymphocytes increased, while the relative proportion of multinucleated spheres decreased. In addition, the peripheral blood monocytes of Sputum−&Urine+ patients were significantly increased compared with Sputum+&Urine− patients. When the neutrophil percentage is less than 72, lymphocyte percentage is more than 9.5, NLR is less than 9.6, or NMLR is less than 2.9, the probability of being diagnosed with GUTB is significantly higher than the probability of being diagnosed with PTB. These results are consistent with our hypothesis that MTBC enters the bloodstream to stimulate the activation of T cells, thereby activating mononuclear phagocytes. The clinical significance of this hypothesis and our subsequent findings is that hematological parameters can be used to distinguish EPTB from other forms of TB.

Interestingly, the proportion of diabetes patients in the Sputum+&Urine+ group is relatively lower than that in the Sputum+&Urine− and Sputum−&Urine− groups (*p* = 0.067 borderline and *p* = 0.001, respectively). Since hyperglycemia can weaken a patient’s immune system, diabetes may cause several infectious diseases [23]. A review indicated that diabetes was associated with an approximately three-fold risk of developing active TB [24]. However, one study found that the odds of developing any form of EPTB was similar in patients with and without diabetes (adjusted odds ratio = 1.04, 95% confidence interval = 0.70−1.56). This suggests that diabetes does not significantly increase the likelihood of EPTB [25], which is consistent with our results. As discussed earlier, participants in group D [S(−)U(−)] were not healthy individuals but were a group of hospital patients who were suspected of having TB. Therefore, the diabetes incidence of group D patients was possibly higher than that of healthy controls.

This study has some limitations. Although we were able to collect data over 10 years, the number of patients with GUTB was relatively small, which may have affected the statistical power. In addition, as the data were obtained from a single medical center, other databases must be used in future studies. We hope that future multicenter studies or meta-analyses can be conducted to prove our results.

## 5. Conclusions

In conclusion, hematology test data, especially the percentage of neutrophils and lymphocytes, and calculated parameters such as the NLR, NMLR, and PLR have the potential to be used for the early detection of GUTB. Although these parameters cannot replace TB culture as the gold standard, they may help distinguish PTB from GUTB. Thus, it may alert clinicians to the possibility of GUTB before the culture is finished processing.

## Figures and Tables

**Figure 1 pathogens-12-00084-f001:**
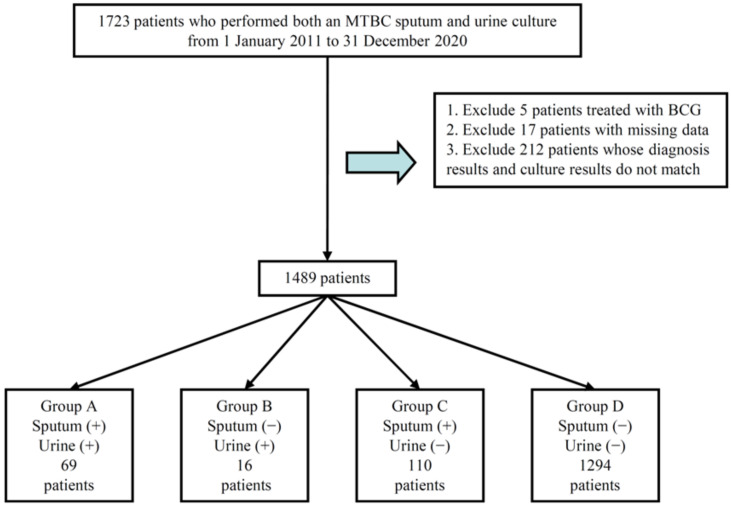
Flow chart of recruitment protocol.

**Figure 2 pathogens-12-00084-f002:**
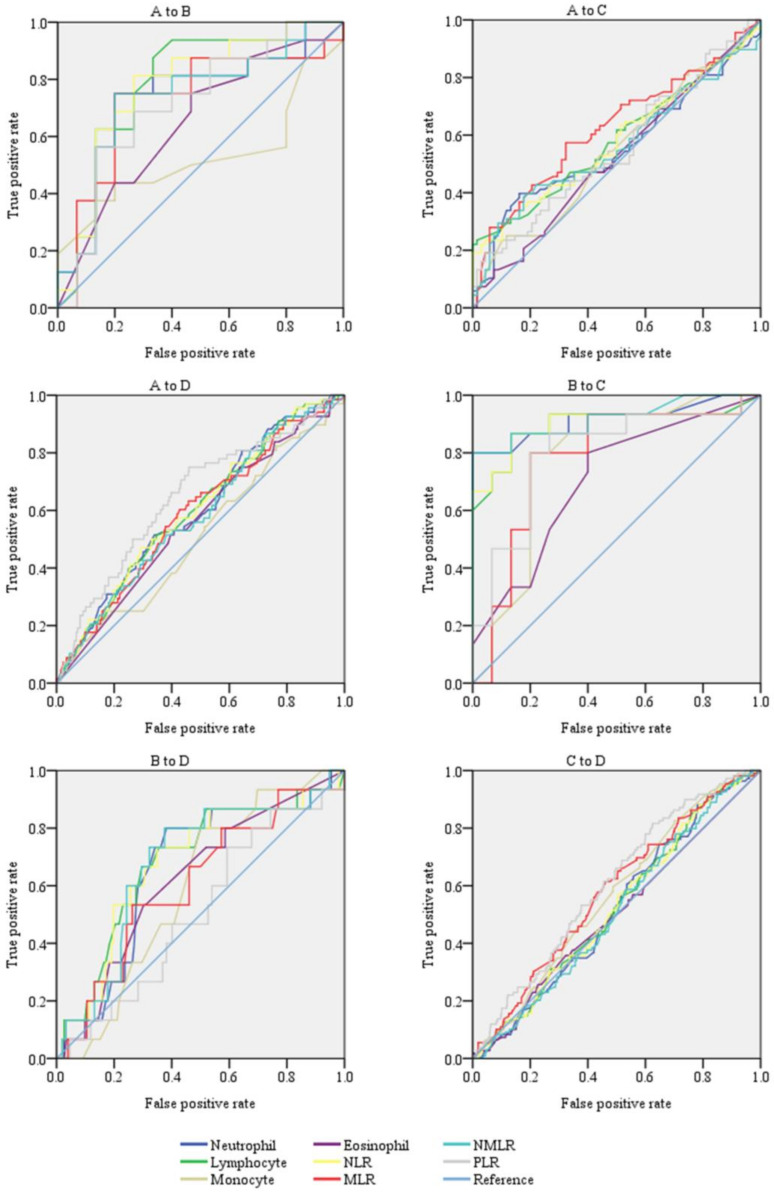
ROC curves for the use of hematological data in the diagnosis of TB.

**Table 1 pathogens-12-00084-t001:** The demographic and clinical outcomes of the 1489 patients.

Category		Group A		Group B		Group C	Group D	*p*-Value ^a^							
		Sputum+&Urine+		Sputum−&Urine+		Sputum+&Urine−	Sputum−&Urine−								
		69		16		110	1294	A to B		A to C	A to D	B to C	B to D		C to D
		*n*	(%)	*n*	(%)	*n*	(%)	*n*	(%)						
Gender	Male	55	(79.7%)	7	(43.8%)	75	(68.2%)	858	(66.3%)						
	Female	14	(20.3%)	9	(56.3%)	35	(31.8%)	436	(33.7%)	0.010 *	0.121	0.025 *	0.089	0.067	0.753
Age (years)	<35	5	(7.2%)	1	(6.3%)	6	(5.5%)	75	(5.8%)						
	≥35, <65	13	(18.8%)	6	(37.5%)	27	(24.5%)	405	(31.3%)						
	≥65	51	(73.9%)	9	(56.3%)	77	(70.0%)	814	(62.9%)	0.243	0.595	0.077	0.447	0.690	0.325
Comorbidities															
CKD	Y	7	(10.1%)	0	(0.0%)	11	(10.0%)	148	(11.4%)						
		62	(89.9%)	16	(100.0%)	99	(90.0%)	1146	(88.6%)	0.338	1.000	0.848	0.356	0.242	0.755
Hypertension	Y	22	(31.9%)	5	(31.3%)	30	(27.3%)	501	(38.7%)						
		47	(68.1%)	11	(68.8%)	80	(72.7%)	793	(61.3%)	1.000	0.612	0.309	0.769	0.615	0.018 *
Diabetes	Y	7	(10.1%)	2	(12.5%)	23	(20.9%)	373	(28.8%)						
		62	(89.9%)	14	(87.5%)	87	(79.1%)	921	(71.2%)	0.675	0.067	0.001 *	0.737	0.263	0.078
IHD	Y	11	(15.9%)	1	(6.3%)	17	(15.5%)	197	(15.2%)						
		58	(84.1%)	15	(93.8%)	93	(84.5%)	1097	(84.8%)	0.449	1.000	1.000	0.465	0.491	1.000
COPD	Y	4	(5.8%)	0	(0.0%)	10	(9.1%)	150	(11.6%)						
		65	(94.2%)	16	(100.0%)	100	(90.9%)	1144	(88.4%)	1.000	0.571	0.172	0.359	0.242	0.445
Liver cirrhosis	Y	6	(8.7%)	0	(0.0%)	3	(2.7%)	53	(4.1%)						
		63	(91.3%)	16	(100.0%)	107	(97.3%)	1241	(95.9%)	0.589	0.090	0.116	1.000	1.000	0.618
Malignancy	Y	13	(18.8%)	2	(12.5%)	22	(20.0%)	379	(29.3%)						
		56	(81.2%)	14	(87.5%)	88	(80.0%)	915	(70.7%)	0.726	1.000	0.075	0.735	0.174	0.047 *
AIDS	Y	3	(4.3%)	0	(0.0%)	0	(0.0%)	62	(4.8%)						
		66	(95.7%)	16	(100.0%)	110	(100.0%)	1232	(95.2%)	1.000	0.056	1.000	N/A	1.000	0.013 *

^a^ The *p*-value was estimated by Student’s *t* test. Abbreviations: CKD = chronic kidney disease, IDH = ischemic heart disease, COPD = chronic obstructive pulmonary disease, AIDS = acquired immune deficiency syndrome. * represents significant values.

**Table 2 pathogens-12-00084-t002:** Laboratory test results between gender and age-matched patients with sputum-TB and urine-TB both positive and patients only with urine-TB.

Categories	A			B			C			D			*p*-Value ^a^		
	Sputum+&Urine+			Sputum−&Urine+			Sputum+&Urine−			Sputum−&Urine−					
	16			16			16			160			A vs. B	B vs. C	B vs. D
	Mean	±	SD	Mean	±	SD	Mean	±	SD	Mean	±	SD			
Glucose (mg/dL)	113.32	±	33.72	97.47	±	22.86	135.38	±	43.97	144.10	±	77.34	0.132	0.006	<0.001
RBC (10^6^/μL)	3.79	±	0.45	4.12	±	0.88	3.55	±	0.80	3.50	±	0.65	0.189	0.065	0.001
Hb (g/dL)	10.69	±	1.76	11.60	±	2.38	10.28	±	2.10	10.46	±	1.90	0.228	0.105	0.028
Hct (%)	32.24	±	5.14	34.93	±	7.07	30.8	±	6.11	31.40	±	5.81	0.228	0.087	0.024
Neutrophil (%)	78.29	±	12.53	67.80	±	13.87	87.69	±	7.92	74.57	±	14.85	0.032	<0.001	0.082
Lymphocyte (%)	11.41	±	7.81	22.29	±	12.48	6.04	±	5.73	15.83	±	12.57	0.006	<0.001	0.052
Monocyte (%)	8.53	±	5.83	7.46	±	2.04	4.85	±	2.96	6.88	±	3.82	0.500	0.007	0.549
NLR	17.3	±	24.3	10.0	±	23.6	32.5	±	27.70	10.7	±	15.5	0.404	0.024	0.875
NMLR	8.2	±	12.1	3.5	±	3.9	12.6	±	7.20	6.3	±	10.3	0.155	<0.001	0.288

^a^ The *p*-value was estimated by Student’s *t* test. Note: Group A and B were matched 1:1 for age and gender by PSM. Group B and C were matched 1:1 for age and gender by PSM. Abbreviations: RBC = red blood cell count, Hb = hemoglobin, Hct = hematocrit, NLR = neutrophil-to-lymphocyte ratio, NMLR = neutrophil-to-monocyte-plus-lymphocyte ratio.

**Table 3 pathogens-12-00084-t003:** Laboratory test results between gender and age-matched patients with sputum-TB and Urine-TB both positive and patients with only urine-TB positive.

Categories	A			C			D			*p* Value ^a^	
	Sputum+&Urine+			Sputum+&Urine−			Sputum−&Urine−				
	69			69			552			A vs. C	A vs. D
	Mean	±	SD	Mean	±	SD	Mean	±	SD		
Glucose (mg/dL)	125.28	±	52.37	155.23	±	72.66	136.85	±	65.83	0.006	0.163
WBC (10^3^/μL)	7.96	±	4.01	9.57	±	4.19	9.58	±	5.39	0.023	0.004
Platelet (10^3^/μL)	210.64	±	124.23	258.48	±	144.14	211.32	±	115.27	0.039	0.964
Neutrophil (%)	80.93	±	11.36	83.07	±	8.04	76.01	±	14.07	0.204	0.006
Lymphocyte (%)	10.31	±	7.68	7.89	±	4.39	13.88	±	10.22	0.025	0.005

^a^ The *p*-value was estimated by Student’s *t* test. Group A and C were matched 1:1 for age and gender by PSM. Group A and D were matched 1:8 for age, gender, and diabetes by PSM. Abbreviations: WBC = white blood cell count.

**Table 4 pathogens-12-00084-t004:** Laboratory test results between gender and age-matched patients with sputum-TB and urine-TB both positive and patients with only urine-TB positive.

Categories	C			D			*p*-Value ^a^
	Sputum+&Urine−			Sputum−&Urine−			
	110			550			
	Mean	±	SD	Mean	±	SD	
RBC (10^6^/μL)	3.75	±	0.80	3.54	±	0.73	0.008
WBC (10^3^/μL)	8.71	±	3.80	9.88	±	6.78	0.012
Platelet (10^3^/μL)	253.43	±	130.59	213.35	±	118.09	0.001
Neutrophil (%)	75.91	±	10.51	73.41	±	15.41	0.038

^a^ The *p*-value was estimated by Student’s *t* test. Group C and D were matched 1:5 for age, gender, malignancy, hypertension, and AIDS by PSM. Abbreviations: RBC = red blood cell count, WBC = white blood cell count.

**Table 5 pathogens-12-00084-t005:** ROC curve analysis for hematological test parameters.

Category	A to B		A to C		A to D		B to C		B to D		C to D	
	AUC	*p*-Value ^a^	AUC	*p*-Value	AUC	*p*-Value	AUC	*p*-Value	AUC	*p*-Value	AUC	*p*-Value
Neutrophil	0.738	0.024 *	0.556	0.258	0.598	0.009 *	0.916	<0.001 *	0.659	0.043 *	0.519	0.531
Lymphocyte	0.790	0.006 *	0.584	0.093	0.600	0.008 *	0.896	<0.001 *	0.682	0.020*	0.525	0.414
Monocyte	0.527	0.797	0.541	0.406	0.511	0.769	0.782	0.008 *	0.591	0.245	0.562	0.040 *
Eosinophil	0.650	0.155	0.521	0.670	0.560	0.108	0.707	0.054	0.623	0.115	0.501	0.970
NLR	0.779	0.008 *	0.422	0.117	0.601	0.007 *	0.898	<0.001 *	0.679	0.023 *	0.525	0.409
MLR	0.725	0.033 *	0.624	0.012 *	0.589	0.018	0.778	0.010 *	0.606	0.176	0.577	0.011 *
NMLR	0.737	0.024 *	0.561	0.221	0.583	0.027 *	0.920	<0.001 *	0.676	0.024 *	0.510	0.734
PLR	0.717	0.040 *	0.556	0.260	0.647	<0.001 *	0.800	0.005 *	0.508	0.920	0.603	0.001 *

^a^ The *p*-value was estimated by ROC curve. Abbreviations: NLR = neutrophil-to-lymphocyte ratio, MLR = monocyte-to-lymphocyte ratio, NMLR = neutrophil-to-monocyte-plus-lymphocyte ratio, PLR = platelet-to-lymphocyte ratio. * represents significant values.

**Table 6 pathogens-12-00084-t006:** Chi-square stratification of the five hematological markers.

Stratification of Category		Sputum−&Urine+		Sputum+&Urine−		*p*-Value ^a^
		*n*	(%)	*n*	(%)	
Neutrophil (%)	>72	4	(25.0%)	16	(100.0%)	<0.001
	≤72	12	(75.0%)	0	(0.0%)	
Lymphocyte (%)	<9.5	3	(18.8%)	14	(87.5%)	<0.001
	≥9.5	13	(81.3%)	2	(12.5%)	
NLR	>9.6	3	(18.8%)	14	(87.5%)	<0.001
	≤9.6	13	(81.3%)	2	(12.5%)	
NMLR	>2.9	4	(25.0%)	16	(100.0%)	<0.001
	≤2.9	12	(75.0%)	0	(0.0%)	
PLR	>293.3	4	(25.0%)	13	(81.3%)	0.004
	≤293.3	12	(75.0%)	3	(18.8%)	

^a^ The *p*-value was estimated by chi-square test. Abbreviation: NLR = neutrophil-to-lymphocyte ratio, NMLR = neutrophil-to-monocyte-plus-lymphocyte ratio, PLR = Platelet-to-lymphocyte ratio.

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
