# Peer review of "Hematological Parameters as Potential Markers for Distinguishing Pulmonary Tuberculosis from Genitourinary Tuberculosis"

_pathogens, 2023, doi:10.3390/pathogens12010084_

Round 1

Reviewer 1 Report

Dear respected editor;

Regarding the article number Pathogens-2102704, titled; Hematological Parameters as Potential Markers for Distinguishing Pulmonary Tuberculosis from Genitourinary Tuberculosis

This study aimed to examine the role of laboratory data in the interpretation of diagnosing GUTB as stated in the abstract.

General points to be considered

-          Title: good  

-         Abstract

-          Abstract: should not include any subsections, so please remove it. : (1) Background, (2) Methods:, (3) Results:, (4) Conclusions:.

1.      Introduction:

Mycobacterium tuberculosis complex (MTBC) is one of the leading infectious agents in Taiwan. In 2019, tuberculosis (TB) in Taiwan was reported 8,732 cases (37 cases per 100,000 population) [1].

-          This should be a single sentence, so full stop should be written after the reference number [1].

-          I did not see any connections between the sentences in the introduction.

-          Author should add some literatures which explain the suspected associations between these hematological parameters with PTB and GUTB.

-          The aim of the study in the abstract and in the introduction parts should be rewritten clearly

2.      Materials and Methods

2.1. Study design, setting and participants

- Authors should add details about the number of participates and details about the place of study.

- Also, selection criteria should be added (inclusion and exclusion criteria).

2.2. Biochemical and hematological data 101

Line 111, the abbreviation of hemoglobin should be (Hb) and not (Hgb).

3.      Results

Line 133-135; spaces should be there. Many grammatical errors in the results and discussion should be corrected. The demographic and clinical outcomes of the four groups [group A: Spu- 133 tum(+)Urine(+), group B: Sputum(-)urine(+), group C: Sputum(+)urine(-), and group D: 134 Sputum(-)urine(-)]are presented in Table 1.

Table 1 and 2 should be improved.

The spaces between the raw 1- 4 should be minimized.

In all tables, all abbreviations must be defined in their full names and must be written under the table.

Statistical values presented in tables should be defined under every table and the ty[e of used test should be mentioned.

4.      Discussion

-          Looks good, but it needs editing because there are many typing errors and grammatical mistakes.

-          Author should try to analysis or evaluate and find the association between the screened parameters with PTB and GUTB and should define which parameters are more significant in diagnosis.

-          Author should elaborate more about the importance of hematology tests as a reference in the diagnosis of which could help to distinguish PTB from GUTB and in the early detection of EPTB, especially GUTB.

5.      Conclusion

Needs improvements.

References:

References need to be prepared according to the journal guidelines. Year of publication must be written in bold.

Author Response

Responses to Reviewer 1 Comments

Abstract

Abstract: should not include any subsections, so please remove it. : (1) Background, (2) Methods:, (3) Results:, (4) Conclusions:.

Response. The subsections have been removed (lines 19, 22, 29, and 36).

  1. Introduction:

Mycobacterium tuberculosis complex (MTBC) is one of the leading infectious agents in Taiwan. In 2019, tuberculosis (TB) in Taiwan was reported 8,732 cases (37 cases per 100,000 population) [1].

-          This should be a single sentence, so full stop should be written after the reference number [1].

Response. Corrected (line 44).

-          I did not see any connections between the sentences in the introduction.

Response. The sentences have been rewritten for a more logical connection (lines 48-49, and 55-59).

-         Author should add some literatures which explain the suspected associations between these hematological parameters with PTB and GUTB.

Response. Added (lines 63-69).

-          The aim of the study in the abstract and in the introduction parts should be rewritten clearly

Response. Thanks for the reviewer suggestion. The aim has been rewritten clearly (lines 21-22, and 74-76).

  1. Materials and Methods

2.1. Study design, setting and participants

- Authors should add details about the number of participates and details about the place of study.

- Also, selection criteria should be added (inclusion and exclusion criteria).

Response: All added including selection criteria (lines 89-103, and Figure 1).

2.2. Biochemical and hematological data 101

Line 111, the abbreviation of hemoglobin should be (Hb) and not (Hgb).

Response: Corrected.

  1. Results

Line 133-135; spaces should be there. Many grammatical errors in the results and discussion should be corrected. The demographic and clinical outcomes of the four groups [group A: Spu- 133 tum(+)Urine(+), group B: Sputum(-)urine(+), group C: Sputum(+)urine(-), and group D: 134 Sputum(-)urine(-)]are presented in Table 1.

Response: Thanks for the reviewer correction. Have been checked and corrected.

-Table 1 and 2 should be improved.

The spaces between the raw 1- 4 should be minimized.

Response: Have been amended.

-In all tables, all abbreviations must be defined in their full names and must be written under the table.

-Statistical values presented in tables should be defined under every table and the ty[e of used test should be mentioned.

Response: Defined all abbreviations and statistical values.

  1. Discussion

-          Looks good, but it needs editing because there are many typing errors and grammatical mistakes.

Response: Have been checked and corrected.

-          Author should try to analysis or evaluate and find the association between the screened parameters with PTB and GUTB and should define which parameters are more significant in diagnosis.

-          Author should elaborate more about the importance of hematology tests as a reference in the diagnosis of which could help to distinguish PTB from GUTB and in the early detection of EPTB, especially GUTB.

Response: Thanks for the reviewer suggestion. We have rewritten some of the Discussion paragraphs (lines 282-291).

  1. Conclusion

Needs improvements.

Response: Thanks for the reviewer suggestion. The statement has been rewritten (lines 310-314).

References:

References need to be prepared according to the journal guidelines. Year of publication must be written in bold.

Response: Already fit the Journal guidelines.

Reviewer 2 Report

Concerning the manuscript; Pathogens-2102704

 Hematological Parameters as Potential Markers for Distinguishing Pulmonary Tuberculosis from Genitourinary Tuberculosis

The paper presented for the review contains an overview of the value of laboratory tests in diagnosing of GUTB.  I have read the manuscript with interest; the manuscript is well written and seems good but minor issues/information should have been included or discussed before the publication:

1.      Line 35, correct the word patents to patients.

2.      In the table 2-5, add measurable unite for all evaluated laboratory test.

Author Response

Responses to Reviewer 2 Comments

  1. Line 35, correct the word patents to patients.

Response. Thanks for the reviewer reminder. We have corrected the mistake.

  1. In the table 2-5, add measurable unite for all evaluated laboratory test.

Response. We have added measurable units for all evaluated laboratory tests.
